# Solving graph compression via optimal transport

**Vikas K. Garg**
CSAIL, MIT
vgarg@csail.mit.edu

**Tommi Jaakkola**
CSAIL, MIT
tommi@csail.mit.edu

## Abstract

We propose a new approach to graph compression by appeal to optimal transport. The transport problem is seeded with prior information about node importance, attributes, and edges in the graph. The transport formulation can be setup for either directed or undirected graphs, and its dual characterization is cast in terms of distributions over the nodes. The compression pertains to the support of node distributions and makes the problem challenging to solve directly. To this end, we introduce Boolean relaxations and specify conditions under which these relaxations are exact. The relaxations admit algorithms with provably fast convergence. Moreover, we provide an exact $O(d \log d)$ algorithm for the subproblem of projecting a $d$-dimensional vector to transformed simplex constraints. Our method outperforms state-of-the-art compression methods on graph classification.

## 1 Introduction

Graphs are widely used to capture complex relational objects, from social interactions to molecular structures. Large, richly connected graphs can be, however, computationally unwieldy if used as-is, and spurious features present in the graphs that are unrelated to the task can be statistically distracting. A significant effort thus has been spent on developing methods for compressing or summarizing graphs towards graph *sketches* [1]. Beyond computational gains, these sketches take center stage in numerous tasks pertaining to graphs such as partitioning [2, 3], unraveling complex or multi-resolution structures [4, 5, 6, 7], obtaining coarse-grained diffusion maps [8], including neural convolutions [9, 10, 11].

State-of-the-art compression methods broadly fall into two categories: (a) sparsification (removing edges) and (b) coarsening (merging vertices). These methods measure spectral similarity between the original graph and a compressed representation in terms of a (inverse) Laplacian quadratic form [12, 13, 14, 15, 16]. Thus, albeit some of these methods approximately preserve the graph spectrum (see, e.g., [1]), they are oblivious to, and thus less effective for, downstream tasks such as classification that rely on labels or attributes of the nodes. Also, the key compression steps in most of these methods are, typically, either heuristic or detached from their original objective [17].

We address these issues by taking a novel perspective that appeals to the theory of optimal transport [18], and develops its connections to minimum cost flow on graphs [19, 20]. Specifically, we interpret graph compression as minimizing the transport cost from a fixed initial distribution supported on all vertices to an unknown target distribution whose size of support is limited by the amount of compression desired. Thus, the compressed graph in our case is a subgraph of the original graph, restricted to a subset of the vertices selected via the associated transport problem. The transport cost depends on the specified prior information such as importance of the nodes and their labels or attributes, and thus can be informed by the downstream task. Moreover, we take into account the underlying geometry toward the transport cost, unlike agnostic measures such as KL-divergence [21].

There are several technical challenges that we must address. First, the standard notion of optimal transport on graphs is tailored to directed graphs where the transport cost decomposes as a directed

flow along the edge orientations [22]. To circumvent this limitation, we extend optimal transport on graphs to handle both directed and undirected edges, and derive a dual that directly measures the discrepancy between distributions on the vertices. As a result, we can also compress *mixed* graphs that contain both directed and undirected edges [23, 24, 25, 26].

The second challenge comes from the compression itself, enforced in our approach as sparse support of the target distribution. Optimal transport (OT) is known to be computationally intensive, and almost all recent applications of OT in machine learning, e.g., [27, 28, 29, 30, 31, 32] rely on entropy regularization [33, 34, 35, 36, 37] for tractability. However, entropy is not conducive to sparse solutions since it discourages the variables from ever becoming zero [22]. In principle, one could consider convex alternatives such as enforcing $\ell_1$ penalty [38]. However, such methods require iterative tuning to find a solution that matches the desired support, and require strong assumptions such as restricted eigenvalue, isometry, or nullspace conditions for recovery. Some of these issues were previously averted by introducing binary selection variables [39, 40] and using Boolean relaxations in the context of unconstrained real spaces (regression). However, they do not apply to our setting since the target distribution must reside in the simplex. We introduce *constrained* Boolean relaxations that are not only efficient, but also provide exactness certificates.

Our graph compression formulation also introduces new algorithmic challenges. For example, solving our sparsity controlled dual transport problem involves a new subproblem of projecting on the probability simplex $\Delta$ under a diagonal transformation. Specifically, let $D(\epsilon)$ be a diagonal matrix with the diagonal $\epsilon \in [0,1]^d \setminus \{\mathbf{0}\}$. Then, for a given $\epsilon$, the problem is to find the projection $x \in \mathbb{R}^d$ of a given vector $y \in \mathbb{R}^d$ such that $D(\epsilon)x \in \Delta$. This generalizes well-studied problem of Euclidean projection on the probability simplex [41, 42, 43], recovered if each $\epsilon_i$ is set to 1. We provide an exact $O(d \log d)$ algorithm for solving this generalized projection. Our approach leads to convex-concave saddle point problems with fast convergence via methods such as Mirror Prox [44].

To summarize, we make the following contributions. We propose an approach for graph compression based on optimal transport (OT). Specifically, we (a) extend OT to undirected and mixed graphs (section 2), (b) introduce constrained Boolean relaxations for our dual OT problem, and provide exactness guarantees (section 3), (c) generalize Euclidean projection onto simplex, and provide an efficient algorithm (section 3), and (d) demonstrate that our algorithm outperforms state-of-the-art compression methods, both in accuracy and compression time, on classifying graphs from standard real datasets. We also provide qualitative results that our approach provides meaningful compression in synthetic and real graphs (section 4).

## 2 Optimal transport for general edges

Let $\vec{G} = (V, \vec{E})$ be a directed graph on nodes (or vertices) $V$ and edges $\vec{E}$. We define the signed incidence matrix $\vec{F}$: $\vec{F}(\vec{e}, v) = 1$ if $\vec{e} = (w, v) \in \vec{E}$ for some $w \in V$, $-1$ if $\vec{e} = (v, w) \in \vec{E}$ for some $w \in V$, and 0 otherwise. Let $c(\vec{e}) \in \mathbb{R}_+$ be the positive cost to transport unit mass along edge $\vec{e} \in \vec{E}$, and $\Delta(V)$ the probability simplex on $V$. The shorthand $a \preceq b$ denotes that $a(i) \leq b(i)$ for each component $i$. Let $\mathbf{0}$ be a vector of all zeros and $\mathbf{1}$ a vector of all ones. Let $\rho_0, \rho_1 \in \Delta(V)$ be distributions over the vertices in $V$. The optimal transport distance $\vec{W}(\rho_0, \rho_1)$ from $\rho_0$ to $\rho_1$ is [22]:

$$\vec{W}(\rho_0, \rho_1) \quad = \quad \min_{\substack{J \in \mathbb{R}^{|\vec{E}|} \\ \mathbf{0} \preceq J}} \quad \sum_{\vec{e} \in \vec{E}} c(\vec{e}) J(\vec{e}) \qquad \text{s.t.} \quad \vec{F}^\top J = \rho_1 - \rho_0 \ ,$$

where $J(\vec{e})$ is the non-negative mass transfer from tail to head on edge $\vec{e}$. Intuitively, $\vec{W}(\rho_0, \rho_1)$ is the minimum cost of a directed flow from $\rho_0$ to $\rho_1$. In order to extend this intuition to the undirected graphs, we need to refine the notion of incidence, and let the mass flow in either direction. Specifically, let $G = (V, E)$ be a connected undirected graph. We define the incidence matrix pertaining to $G$ as $F(e, v) = 1$ if edge $e$ is incident on $v$, and 0 otherwise. With each undirected edge $e \in E$, having cost $c(e) \in \mathbb{R}_+$, we associate two directed edges $e^+$ and $e^-$, each with cost $c(e)$, and flow variables $J^+(e), J^-(e) \geq 0$. Then, the total *undirected flow* pertaining to $e$ is $J^+(e) + J^-(e)$. Since we incur cost for flow in either direction, we define the optimal transport cost $W(\rho_0, \rho_1)$ from $\rho_0$ to $\rho_1$ as

$$\min_{\substack{J^+, J^- \in \mathbb{R}^{|E|} \\ \mathbf{0} \preceq J^+, J^-}} \quad \sum_{e \in E} c(e)(J^+(e) + J^-(e)) \qquad \text{s.t.} \quad F^\top(J^- - J^+) = \rho_1 - \rho_0 \ . \tag{1}$$

We call a directed edge $e^+$ *active* if $J^+(e) > 0$, i.e., there is some positive flow on the edge (likewise for $e^-$). Moreover, by extension, we call an undirected edge $e$ active if at least one of $e^+$ and $e^-$ is active. We claim that at most one of $e^+$ and $e^-$ may be active for any edge $e$.

**Theorem 1.** *The optimal solution to* (1) *must have* $J^+(e) = 0$ *or* $J^-(e) = 0$ *(or both)* $\forall\, e \in E$.

The proof is provided in the supplementary material. Thus, like the directed graph setting, we either have flow in only one direction for each edge $e$, or no flow at all. Moreover, Theorem 1 facilitates generalizing optimal transport distance to mixed graphs $\tilde{G}(V, E, \vec{E})$, i.e., where both directed and undirected edges may be present. In particular, we adapt the formulation in (1) with minor changes: (a) we associate bidirectional variables with each edge, directed or undirected. For the undirected edges $e \in E$, we replicate the constraints from (1). For the directed edges $\vec{e}$, we follow the convention that $J^+(\vec{e})$ denotes the outward flow along $\vec{e}$ whereas $J^-(\vec{e})$ denotes the incoming flow (from head to tail), and impose the additional constraints $J^-(\vec{e}) = 0$. We will focus on undirected graphs $G = (V, E)$ since the extensions to directed and mixed graphs are immediate due to Theorem 1.

## 3 Graph compression

We view graph compression as the problem of minimizing the optimal transport distance from an initial distribution $\rho_0$ having full support on the vertices $V$ to a target distribution $\rho_1$ that is supported only on a subset $S_V(\rho_1)$ of $V$. The compressed subgraph is obtained by restricting the original graph to vertices in $S_V(\rho_1)$ and the incident edges. The initial distribution $\rho_0$ encodes any prior information. For instance, it might be taken as a stationary distribution of random walk on the graph. Likewise, the cost function $c$ encodes the preference for different edges. In particular, a high value of $c(e)$ would inhibit edge $e$ from being active. This flexibility allows our framework to inform compression based on the specifics of different downstream applications by getting to define $\rho_0$ and $c$ appropriately.

### 3.1 Dual characterization of the transport distance

Note that (1) defines an optimization problem over edges. However, our perspective requires quantifying $W(\rho_0, \rho_1)$ as an optimization over the vertices. Fortunately, strong duality comes to our rescue. Let $c = (c(e), e \in E)$ be the column vector obtained by stacking the costs. The dual of (1) is

$$\max_{\substack{\mathbf{0} \preceq y, \mathbf{0} \preceq z \\ -c \preceq F(y-z) \preceq c}} (y-z)^\top (\rho_1 - \rho_0), \quad \text{or equivalently,} \quad \max_{\substack{t \in \mathbb{R}^{|V|} \\ -c \preceq Ft \preceq c}} t^\top (\rho_1 - \rho_0) \ . \tag{2}$$

This alternative formulation of $W(\rho_0, \rho_1)$ in (2) lets us define compression solely in terms of variables over vertices. Specifically, for a budget of at most $k$ vertices, we solve

$$\min_{\substack{\rho_1 \in \Delta(V) \\ ||\rho_1||_0 \leq k}} \max_{\substack{t \in \mathbb{R}^{|V|} \\ -c \preceq Ft \preceq c}} \underbrace{t^\top (\rho_1 - \rho_0) + \frac{\lambda}{2}||\rho_1||^2}_{\mathcal{L}_\lambda(\rho_1, t; \rho_0)} \ , \tag{3}$$

where $\lambda > 0$ is a regularization hyperparameter, and $||\rho_1||_0$ is the number of vertices with positive mass under $\rho_1$, i.e., the cardinality of support set $S_V(\rho_1)$. The quadratic penalty is strongly convex in $\rho_1$, so as we shall see shortly, would help us leverage fast algorithms for a saddle point problem. Note that a high value of $\lambda$ would encourage $\rho_1$ toward a uniform distribution. We favor this penalty over entropy, which is not conducive to sparse solutions since entropy would forbid $\rho_1$ from having zero mass at any vertex in the graph.

Our next result reveals the structure of optimal solution $\rho_1^*$ in (3). Specifically, $\rho_1^*$ must be expressible as an affine function of $\rho_0$ and $F$. Moreover, the constraints on active edges are tight. This reaffirms our intuition that $\rho_1^*$ is obtained from $\rho_0$ by transporting mass along a subset of the edges, i.e., the active edges. The remaining edges do not participate in the flow.

**Theorem 2.** *The optimal $\rho_1^*$ in* (3) *is of the form* $\rho_1^* = \rho_0 + F^\top \eta$ , *where* $\eta \in \mathbb{R}^{|E|}$. *Furthermore, for any active edge* $e \in E$, *we must have* $Ft^*(e) \in \{c(e), -c(e)\}$.

### 3.2 Constrained Boolean relaxations

The formulation (3) is non-convex due to the support constraint on $\rho_1$. Since recovery under $\ell_1$ based methods such as Lasso often requires extensive tuning, we resort to the method of Boolean

relaxations that affords an explicit control much like the $\ell_0$ penalty. However, prior literature on Boolean relaxations is limited to variables that have no additional constraints beyond sparsity. Thus, in order to deal with the simplex constraints $\rho_1 \in \Delta(V)$, we introduce constrained Boolean relaxations. Specifically, we define the characteristic function $g_V(x) = 0$ if $x \in \Delta(V)$ and $\infty$ otherwise, and move the non-sparsity constraints inside the objective. This lets us delegate the sparsity constraints to binary variables, which can be relaxed to $[0,1]$. Using the definition of $\mathcal{L}_\lambda$, we can write (3) as

$$\min_{\substack{\rho_1 \in \mathbb{R}^{|V|} \\ ||\rho_1||_0 \le k}} \max_{\substack{t \in \mathbb{R}^{|V|} \\ -c \preceq Ft \preceq c}} \mathcal{L}_\lambda(\rho_1, t; \rho_0) + g_V(\rho_1) .$$

Denoting by $\odot$ the Hadamard (elementwise) product, and introducing variables $\epsilon \in \{0,1\}^{|V|}$, we get

$$\min_{\substack{\epsilon \in \{0,1\}^{|V|} \\ ||\epsilon||_0 \le k}} \min_{\rho_1 \in \mathbb{R}^{|V|}} \max_{\substack{t \in \mathbb{R}^{|V|} \\ -c \preceq Ft \preceq c}} \mathcal{L}_\lambda(\rho_1 \odot \epsilon, t; \rho_0) + g_V(\rho_1 \odot \epsilon) .$$

Adjusting the characteristic term as a constraint, we have the following equivalent problem

$$\min_{\substack{\epsilon \in \{0,1\}^{|V|} \\ ||\epsilon||_0 \le k}} \min_{\substack{\rho_1 \in \mathbb{R}^{|V|} \\ \rho_1 \odot \epsilon \in \Delta(V)}} \max_{\substack{t \in \mathbb{R}^{|V|} \\ -c \preceq Ft \preceq c}} \mathcal{L}_\lambda(\rho_1 \odot \epsilon, t; \rho_0) . \tag{4}$$

---

**Algorithm 1** Algorithm to compute Euclidean projection on the $d$-simplex $\Delta$ under a diagonal transformation.

---

**Input:** $y, \epsilon$
Define $\mathcal{I}_> \triangleq \{j \in [d] \mid \epsilon_j > 0\}$
Define $\mathcal{I}_= \triangleq \{j \in [d] \mid \epsilon_j = 0\}$;
$y_> \triangleq \{y_j \mid j \in \mathcal{I}_>\}$; $\epsilon_> \triangleq \{\epsilon_j \mid j \in \mathcal{I}_>\}$
Sort $y_>$ into $\hat{y}_>$ and $\epsilon_>$ into $\hat{\epsilon}_>$, in non-increasing order, based on $y_j/\epsilon_j, j \in \mathcal{I}_>$ . Rename indices in $(\hat{y}_>, \hat{\epsilon}_>)$ to start from 1. Let $\pi$ map $j \in \mathcal{I}_>$ to $\pi(j) \in [|\hat{y}_>|]$. Thus

$$\hat{y}_1/\hat{\epsilon}_1 \ge \hat{y}_2/\hat{\epsilon}_2 \ge \ldots \ge \hat{y}_{|\mathcal{I}_>|}/\hat{\epsilon}_{|\mathcal{I}_>|}$$

$$b_j = \hat{y}_j + \hat{\epsilon}_j \frac{(1 - \sum_{i=1}^j \hat{\epsilon}_i \hat{y}_i)}{\sum_{i=1}^j \hat{\epsilon}_i^2}, \ \forall j \in [|y_>|]$$
$$\ell = \max\{j \in [|y_>|] \mid b_j > 0\}$$
$$\alpha = \frac{(1 - \sum_{i=1}^\ell \hat{\epsilon}_i \hat{y}_i)}{\sum_{i=1}^\ell \hat{\epsilon}_i^2}$$
$$x_j = \max\{\hat{y}_{\pi(j)} + \alpha \hat{\epsilon}_{\pi(j)}, 0\}, \ \forall j \in \mathcal{I}_>$$
$$x_j = y_j, \forall j \in \mathcal{I}_=$$

---

**Algorithm 2** Mirror Prox algorithm to (approximately) find $\epsilon$ in relaxation of (9). The step-sizes at time $\ell$ with respect to $\epsilon$, $t$, and $\zeta$ are $\alpha_\ell$, $\beta_\ell$, and $\gamma_\ell$ respectively.

---

**Input:** $\rho_0, k, \lambda$; iterations $T$
Define $\tilde{\mathcal{E}}_k = \{\epsilon \in [0,1]^{|V|} \mid \epsilon^\top \mathbf{1} \le k\}$
Define $\mathcal{T}_{F,c}$ as in (5)
Define $\psi_{\rho_0}(\epsilon, t, \zeta)$ as in (9)
Initialize $\epsilon^{(0)} = k\mathbf{1}/|V|$, $t^{(0)} = \mathbf{0}$, and $\zeta^{(0)} = 0$
**for** $\ell = 0, 1, \ldots, T$ **do**
    Gradient step:
$$\hat{\epsilon}^{(\ell)} = \mathrm{Proj}_{\tilde{\mathcal{E}}_k}\left(\epsilon^{(\ell)} - \alpha_\ell \nabla_\epsilon \psi_{\rho_0}(\epsilon^{(\ell)}, t^{(\ell)}, \zeta^{(\ell)})\right)$$
$$\hat{t}^{(\ell)} = \mathrm{Proj}_{\mathcal{T}_{F,c}}\left(t^{(\ell)} + \beta_\ell \nabla_t \psi_{\rho_0}(\epsilon^{(\ell)}, t^{(\ell)}, \zeta^{(\ell)})\right)$$
$$\hat{\zeta}^{(\ell)} = \zeta^{(\ell)} + \gamma_\ell \nabla_\zeta \psi_{\rho_0}(\epsilon^{(\ell)}, t^{(\ell)}, \zeta^{(\ell)})$$
    Extra-gradient step:
$$\epsilon^{(\ell+1)} = \mathrm{Proj}_{\tilde{\mathcal{E}}_k}\left(\epsilon^{(\ell)} - \alpha_\ell \nabla_\epsilon \psi_{\rho_0}(\hat{\epsilon}^{(\ell)}, \hat{t}^{(\ell)}, \hat{\zeta}^{(\ell)})\right)$$
$$t^{(\ell+1)} = \mathrm{Proj}_{\mathcal{T}_{F,c}}\left(t^{(\ell)} + \beta_\ell \nabla_t \psi_{\rho_0}(\hat{\epsilon}^{(\ell)}, \hat{t}^{(\ell)}, \hat{\zeta}^{(\ell)})\right)$$
$$\zeta^{(\ell+1)} = \zeta^{(\ell)} + \gamma_\ell \nabla_\zeta \psi_{\rho_0}(\hat{\epsilon}^{(\ell)}, \hat{t}^{(\ell)}, \hat{\zeta}^{(\ell)})$$
**end for**
$$\hat{\epsilon} = \sum_{\ell=1}^T \alpha_\ell \hat{\epsilon}^{(\ell)} / \sum_{\ell=1}^T \alpha_\ell$$

---

Our formulation in (4) requires solving a new subproblem, namely, Euclidean projection on the $d$-simplex $\Delta$ under a diagonal transformation. Specifically, let $D(\epsilon)$ be a diagonal matrix with the diagonal $\epsilon \in [0,1]^d \setminus \{\mathbf{0}\}$. Then, for a given $\epsilon$, the problem is to find the projection $x \in \mathbb{R}^d$ of a given vector $y \in \mathbb{R}^d$ such that $D(\epsilon)x = x \odot \epsilon \in \Delta$. This problem generalizes Euclidean projection on the probability simplex [41, 42, 43], which is recovered when we set $\epsilon$ to $\mathbf{1}$, i.e., an all-ones vector. Our next result shows that Algorithm 1 solves this problem exactly in $O(d \log d)$ time.

**Theorem 3.** *Let $\epsilon \in [0,1]^d \setminus \{\mathbf{0}\}$ be a given vector of weights, and $y \in \mathbb{R}^d$ be a given vector of values. Algorithm 1 solves the following problem in $O(d \log d)$ time*

$$\min_{x \in \mathbb{R}^d \,:\, x \odot \epsilon \in \Delta} \frac{1}{2}||x - y||^2 .$$

Theorem 3 allows us to relax the problem (4), and solve the relaxed problem efficiently since the projection steps on the other constraint sets can be solved by known methods [45, 46]. Specifically, since $\epsilon$ consists of only zeros and ones, $||\epsilon||_0 = ||\epsilon||_1 = \epsilon^\top \mathbf{1}$ and $\epsilon \odot \epsilon = \epsilon$. So, we can write (4) as

$$\min_{\substack{\epsilon \in \mathcal{E}_k \\ }} \min_{\substack{\rho_1 \in \mathbb{R}^{|V|} \\ \rho_1 \odot \epsilon \in \Delta(V)}} \max_{t \in \mathcal{T}_{F,c}} \mathcal{L}_\lambda(\rho_1 \odot \epsilon \odot \epsilon, t; \rho_0) ,$$

where we denote the constraints for $\epsilon$ and $t$ respectively by

$$\mathcal{E}_k \triangleq \left\{\epsilon \in \{0,1\}^{|V|} \mid \epsilon^\top \mathbf{1} \leq k\right\}, \quad \text{and} \quad \mathcal{T}_{F,c} \triangleq \left\{t \in \mathbb{R}^{|V|} \mid -c \preceq Ft \preceq c\right\}. \tag{5}$$

We can thus eliminate $\epsilon$ from the regularization term via a change of variable $\rho_1 \odot \epsilon \to \tilde{\rho}_1$

$$\min_{\epsilon \in \mathcal{E}_k} \min_{\substack{\tilde{\rho}_1 \in \mathbb{R}^{|V|} \\ \tilde{\rho}_1 \odot \epsilon \in \Delta(V)}} \max_{t \in \mathcal{T}_{F,c}} \mathcal{L}_0(\tilde{\rho}_1 \odot \epsilon, t; \rho_0) + \frac{\lambda}{2}||\tilde{\rho}_1||^2 . \tag{6}$$

We note that (6) is a mixed-integer program due to constraints $\mathcal{E}_k$, and thus hard to solve. Nonetheless, we can relax the hard binary constraints on the coordinates of $\epsilon$ to $[0, 1]$ intervals to obtain a saddle point formulation with a strongly convex term, and solve the relaxation efficiently, e.g., via customized versions of methods such as Mirror Prox [44], Accelerated Gradient Descent [47], or Primal-Dual Hybrid Gradient [48]. An attractive property of our relaxation is that if the solution $\hat{\epsilon}$ from the relaxed problem is integral then $\hat{\epsilon}$ must be optimal for the non-relaxed hard problem (6), and so the original formulation (3). We now pin down the necessary and sufficient conditions for optimality of $\hat{\epsilon}$.

**Theorem 4.** *Let $S_V(\rho_1^*) = \{v \in V \mid \rho_1^*(v) > 0\}$ be the support of optimal $\rho_1$ in the original formulation* (3)*. Let the indicator $I_{S_V^*} \in \{0,1\}^{|V|}$ be such that $I_{S_V^*}(v) = 1$ if $v \in S_V(\rho_1^*)$ and $0$ otherwise. The relaxation of* (6) *is guaranteed to recover $S_V(\rho_1^*)$ if and only if there exists a tuple $(\gamma, \hat{t}, \hat{\nu}, \hat{\zeta}) \in \mathbb{R}_+ \times \mathbb{R}^{|V|} \times \mathbb{R}_+^{|V|} \times \mathbb{R}$ such that the following holds for all vertices $v \in V$,*

$$|\hat{t}(v) - \hat{\nu}(v) + \hat{\zeta}| \quad \begin{cases} > \gamma & \text{if } v \in S_V(\rho_1^*) \\ < \gamma & \text{if } v \notin S_V(\rho_1^*) \end{cases} , \quad \text{where} \tag{7}$$

$$(\hat{t}, \hat{\nu}, \hat{\zeta}) \in \arg\max_{t \in \mathcal{T}_{F,c}} \max_{\nu \in \mathbb{R}_+^{|V|}} \max_{\zeta \in \mathbb{R}} -\left(\frac{1}{2\lambda}||(t - \nu + \zeta\mathbf{1}) \odot I_{S_V^*}||^2 + t^\top \rho_0 + \zeta\right). \tag{8}$$

The quantity $|\hat{t}(v) - \hat{\nu}(v) + \hat{\zeta}|$ in (7) may be viewed as the strength of a signal. In that sense, we require the vertices in support of optimal $\rho_1$ to have a strictly higher signal that the vertices not in the support. Such signal detection conditions appear in various contexts and often have information theoretic implications, e.g., Ising models [49]. (7) is also reminiscent of the $\beta$-min condition on regression coefficients for variable selection with Lasso in high-dimensional linear models [50].

For some applications projecting on the simplex, as required by (6), may be an expensive operation. We can invoke the minimax theorem to swap the order of $\tilde{\rho}_1$ and $t$, and proceed with a Lagrangian dual to eliminate $\tilde{\rho}_1$ at the expense of introducing a scalar variable. Thus, effectively, we can replace the projection on simplex by a one-dimensional search. We state this equivalent formulation below.

**Theorem 5.** *Problem* (6)*, and thus the original formulation* (3)*, is equivalent to*

$$\min_{\epsilon \in \mathcal{E}_k} \max_{\substack{t \in \mathcal{T}_{F,c} \\ \zeta \in \mathbb{R}}} \underbrace{-\frac{1}{2\lambda} \sum_{v:t(v) \leq -\zeta} \left(\epsilon(v)(t(v) + \zeta)^2 + 2\lambda t(v)\rho_0(v)\right) - \sum_{v:t(v) > -\zeta} t(v)\rho_0(v) - \zeta}_{\psi_{\rho_0}(\epsilon, t, \zeta)} . \tag{9}$$

We present a customized Mirror Prox procedure in Algorithm 2. The projections $\text{Proj}_{\mathcal{T}_{F,c}}$ and $\text{Proj}_{\bar{\mathcal{E}}_k}$ can be computed efficiently [45, 46]. We round the solution $\hat{\epsilon} \in [0, 1]^{|V|}$ returned by the algorithm to have at most $k$ vertices as the estimated support for the target distribution $\rho_1$ if $\hat{\epsilon}$ is not integral. The compressed graph is taken to be the subgraph spanned by these vertices.

### 3.3 Specifying the cost function

In our experiments, we fixed the cost of each edge, computed based on the agreement between the associated vertex labels. Here we illustrate briefly how to parameterize the cost and how the parameters could be learned. Define $\ell(i, j) = 1$ for edge $(i, j)$ if vertices $i$ and $j$ have the same label, and $-1$ otherwise. Let the cost function be parameterized by $\theta = (\theta_s, \theta_d), \theta_s > 0, \theta_d > 0$ such that

$$c_\theta(i, j) = 0.5(\theta_d(1 - \ell(i, j)) + \theta_s(1 + \ell(i, j))) .$$

Table 1: **Description of graph datasets, and comparison of accuracy on test data.** We provide the statistics on the number of graphs, number of classes, average number of nodes, and average number of edges in each dataset. The classification test accuracy (along with standard deviation) when each graph was (roughly) compressed to half is shown for each method for each training fraction in $\{0.2, \ldots, 0.8\}$. The algorithm having the best performance is indicated with bold font in each case. '-' entries indicate that the method failed to compress the dataset (e.g. due to matrix singularity).

| Dataset | method | acc@0.2 | acc@0.3 | acc@0.4 | acc@0.5 | acc@0.6 | acc@0.7 | acc@0.8 |
|---|---|---|---|---|---|---|---|---|
| **MSRC-21C** | REC | .485±.016 | .543±.010 | .595±.010 | .625±.013 | .641±.008 | .696±.013 | .738±.016 |
| graphs: 209 | Heavy | .408±.016 | .479±.015 | .516±.009 | .538±.009 | .557±.011 | .602±.022 | .653±.011 |
| classes: 20 | Affinity | .413±.021 | .489±.008 | .516±.011 | .549±.010 | .560±.016 | .607±.019 | .654±.021 |
| nodes: 40.3 | Alg. Dist. | .452±.036 | .498±.035 | .524±.021 | .535±.027 | .531±.029 | .590±.032 | .652±.044 |
| edges: 96.6 | OTC | **.548±.004** | **.605±.003** | **.639±.006** | **.679±.003** | **.696±.002** | **.742±.007** | **.778±.005** |
| **DHFR** | REC | .681±.011 | .704±.014 | .724±.007 | .738±.009 | .749±.008 | .756±.011 | .771±.011 |
| graphs: 467 | Heavy | .719±.010 | .751±.010 | .776±.012 | .782±.008 | .777±.009 | .786±.014 | .799±.013 |
| classes: 2 | Affinity | .717±.013 | .733±.011 | .745±.014 | .761±.014 | .771±.019 | .767±.015 | .785±.013 |
| nodes: 42.4 | Alg. Dist. | .743±.011 | .761±.012 | .768±.022 | .786±.019 | .810±.025 | **.817±.033** | .809±.030 |
| edges: 44.5 | OTC | **.757±.004** | **.784±.003** | **.797±.005** | **.799±.003** | **.811±.007** | .814±.006 | **.823±.004** |
| **MSRC-9** | REC | .738±.011 | .782±.010 | .817±.009 | .818±.013 | .835±.020 | .833±.018 | .840±.013 |
| graphs: 221 | Heavy | .648±.019 | .710±.024 | .766±.014 | .773±.010 | .786±.009 | .796±.010 | .813±.009 |
| classes: 8 | Affinity | .665±.015 | .722±.005 | .762±.010 | .774±.014 | .789±.026 | .786±.019 | .801±.017 |
| nodes: 40.6 | Alg. Dist. | .666±.048 | .717±.051 | .756±.029 | .771±.039 | .798±.032 | .803±.030 | .809±.046 |
| edges: 97.9 | OTC | **.784±.005** | **.808±.005** | **.826±.007** | **.846±.003** | **.839±.006** | **.842±.007** | **.854±.003** |
| **BZR-MD** | REC | .525±.011 | .548±.015 | **.563±.020** | .553±.021 | .563±.012 | .569±.012 | .587±.020 |
| graphs: 306 | Heavy | .497±.000 | .546±.000 | .555±.000 | .522±.000 | .550±.000 | .572±.000 | .558±.000 |
| classes: 2 | Affinity | .508±.006 | .534±.012 | .534±.017 | .532±.015 | .549±.020 | .567±.033 | .562±.029 |
| nodes: 21.3 | Alg. Dist. | .497±.021 | .546±.026 | .555±.038 | .522±.028 | .550±.028 | .572±.024 | .558±.039 |
| edges: 225.06 | OTC | **.534±.000** | **.569±.000** | .547±.000 | **.579±.000** | **.572±.000** | **.607±.000** | **.603±.000** |
| **Mutagenicity** | REC | .713±.006 | .730±.006 | .742±.005 | .752±.004 | .758±.005 | .765±.007 | .769±.007 |
| graphs: 4337 | Heavy | .718 ±.006 | .738±.004 | .753±.004 | .763±.004 | .771±.003 | .779±.004 | .783±.004 |
| classes: 2 | Affinity | - | - | - | - | - | - | - |
| nodes: 30.3 | Algebraic | - | - | - | - | - | - | - |
| edges: 30.8 | OTC | **.749±.002** | **.768±.003** | **.779±.003** | **.787±.004** | **.792±.004** | **.795±.003** | **.799±.003** |

Thus, $c_\theta(i, j) \in \{\theta_s, \theta_d\}$ depending on whether $i$ and $j$ have the same label.

The cost parameters couldn't be driven solely by the compression criterion as this objective would lead to a trivial all-zero solution. Instead, $\theta$ must be in part driven by an external classification loss. In other words, we can learn $\theta$ by trading off compression loss against, for example, the ability to correctly classify the resulting reduced graphs. We leave this for future work.

### 3.4 Relation to other compression techniques

Most compression algorithms try to preserve the graph spectrum via a (multi-level) coarsening procedure: at each level they compute a matching of vertices and merge the matched vertices, e.g., Heavy Edge [2] contracts those edges $(i, j)$ that are incident on low degree vertices. Likewise REC [1] follows a randomized greedy procedure for generating maximal matching incrementally. Let $d_i$ be the degree of node $i$. Setting the cost $c(i, j) = \max(d_i, d_j)$ in our framework will incentivize flow on edges with low degree vertices, and in turn, compression of one of their end points. The vertices not in support of target distribution may then be viewed as being matched to (a subset of) adjacent vertices that they transfer flow to. Unlike other methods, our approach is flexible in terms of defining $c(i, j)$.

## 4 Experiments

We conducted several experiments to demonstrate the merits of our method. We start by describing the experimental setup. We fixed the value of hyperparameters in Algorithm 2 for all our experiments. Specifically, we set the regularization coefficient $\lambda = 1$, and the gradient rates $\alpha_\ell = 0.1, \beta_\ell =$

$0.1, \gamma_\ell = 0.1$ for each $\ell \in \{0, 1, \ldots, T\}$. We also let $\rho_0$ be the stationary distribution by setting $\rho_0(v)$ for each $v \in V$ as the ratio of $deg(v)$, i.e. the degree of $v$, to the sum of degrees of all the vertices. Note that the distribution thus obtained is the unique stationary distribution for connected non-bipartite graphs, and *a* stationary distribution for the bipartite graphs. Moreover, for non-bipartite graphs, it has a nice physical interpretation in that any random walk on the graph always converges to this distribution irrespective of the graph structure.

The objective of our experiments is three-fold. Since compression is often employed as a preprocessing step for further tasks, we first show that our method compares favorably, in terms of test accuracy, to the state-of-the-art compression methods on graph classification. We then demonstrate that our method performed the best in terms of the compression time. We finally show that our approach provides qualitatively meaningful compression on synthetic and real examples.

## 4.1 Classifying standard graph data

We used several standard graph datasets for our experiments, namely, DHFR [51], BZR-MD [52], MSRC-9, MSRC-21C [53], and Mutagenicity [54]. We focused on these datasets since they represent a wide spectrum in terms of the number of graphs, number of classes, average number of nodes per graph, and average number of edges per graph (see Table 1 for details). All these datasets have a class label for each graph, and additionally, labels for each node in every graph.

We compare the test accuracy of our algorithm, OTC (short for Optimal Transport based Compression), to several state-of-the-art methods: REC [1], Heavy edge matching (Heavy) [2], Affinity vertex proximity (Affinity) [14], and Algebraic distance (Algebraic) [12, 13]. Amongst these, the REC algorithm is a randomized method that iteratively contracts edges of the graph and thereby coarsens the graph. Since the method is randomized, REC yields a compressed graph that achieves a specified compression factor, i.e., the ratio of the number of nodes in the reduced graph to that in the original uncompressed graph, only in expectation. Therefore, in order to allow for a fair comparison, we first run REC on each graph with the compression factor set to 0.5, and then execute other baselines and our algorithm, i.e. Algorithm 2, with $k$ set to the number of nodes in the compressed graph produced by REC. Also to mitigate the effects of randomness on the number of nodes returned by REC, we performed 5 independent runs for REC, and subsequently all other methods.

Each collection of compressed graphs was then divided into train and test sets, and used for our classification task. Since our datasets do not provide separate train and test sets, we employed the following procedure for each dataset. We partitioned each dataset into multiple train and test sets of varying sizes. Specifically, for each $p \in \{0.2, 0.3, 0.4, 0.5, 0.6, 0.7, 0.8\}$, we divided each dataset randomly into a train set containing a fraction $p$ of the graphs in the dataset, and a test set containing the remaining $1 - p$ fraction. To mitigate the effects of chance, we formed 5 such independent train-test partitions for each fraction $p$ for each dataset. We averaged the results over these multiple splits to get one reading per collection, and thus 5 readings in total for all collections, for each fraction for each method. We averaged the test accuracy across collections for each method and fraction.

We now specify the cost function $c$ for our algorithm. As described in section 3, we can leverage the cost function to encode preference for different edges in the compressed graph. For each graph, we set $c = 0.01$ for each edge $e$ incident on the nodes with same label, and $c = 0.02$ for each $e$ incident on the nodes that have different label. Thus, in effect, we slightly biased the graphs to prefer retaining the edges that have separate labels at their end-points. In our experiments, we employed support vector machines (SVMs), with Weisfeiler-Leman subtree kernel [55] to quantify the similarity between graphs [53, 56, 57]. This kernel is based on the Weisfeiler-Leman test of isomorphism [58, 59], and thus naturally takes into account the labels of the nodes in the two graphs. We fixed the number of kernel iterations to 5. We also fixed $T = 25$ for our algorithm. For each method and each train-test split, we used a separate 5-fold cross-validation procedure to tune the coefficient of error term $C$ over the set $\{0.1, 1, 10\}$ for training an independent SVM model on the training portion.

Table 1 summarizes the performance of different methods for each fraction of the data. As the numbers in bold indicate, our method generally outperformed the other methods across the datasets. We observe that on some datasets the discrepancy between the average test accuracy of two algorithms is massive for every fraction. Note that though OTC performs the best on DHFR, the performance of most methods is similar (except REC, which lags behind). In contrast, REC performs better than all methods except OTC on MSRC-9. This seems to suggest that REC performs well on graphs with

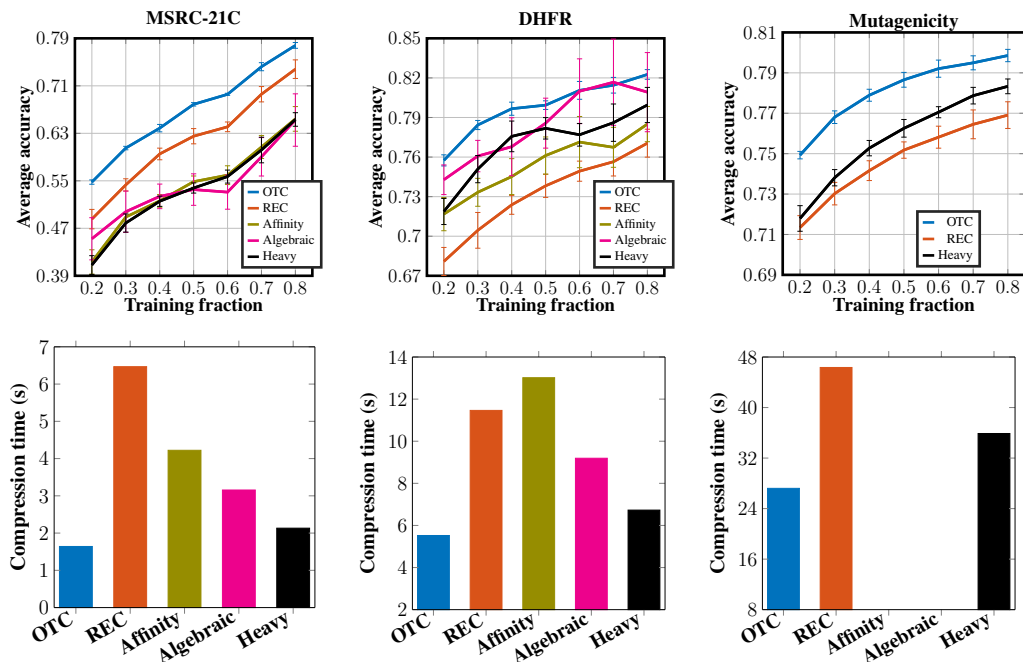

Figure 1: **Comparison on standard graph datasets**. The top row shows the average test accuracy and corresponding standard deviation for our method (OTC) and state-of-the-art baselines for different fractions of training data. The bottom row compares the corresponding compression times. Our method outperforms the other methods in terms of both accuracy and compression time.

strong connectivity, while others might be better on data with a long backbone besides these ring structures. We believe robust performance of OTC across these datasets comprising graphs with vastly different topologies underscores the promise of our approach.

Further, as Fig. 1 shows, OTC performed best in terms of compression time as well. We emphasize that the discrepancy in compression times became quite stark for larger datasets (i.e., DHFR and Mutagenicity). To provide more evidence on the scalability of our method, we also experimented with the larger Tox21AR-LBD data,[1] which consists of about 8600 graphs. Both our method and Algebraic distance performed very well in terms of classification accuracy ($\sim 97\%$) on this data. Our approach took in total about 39 seconds to compress graphs in this dataset to 90% (low compression), and about 41 seconds in total to compress to 10% (high compression). In contrast, the Algebraic distance method took about 48 seconds to compress to 90% and a significantly longer time, i.e., 3.5 minutes to compress to 10%. The other baselines failed to compress this data.

## 4.2 Compressing synthetic and real examples

We now describe our second set of experiments with both synthetic and real data to show that our method can be seeded with useful prior information toward preserving interesting patterns in the compressed structures. This flexibility in specifying the prior information makes our approach especially well-suited for downstream tasks, where domain knowledge is often available.

Fig. 2 demonstrates the effect of compressing a synthetic tree-structured graph. The penultimate level consists of four nodes, each of which has four child nodes as leaves. We introduce asymmetry with respect to the different nodes by specifying different combinations of $c(e)$ for edges $e$ between the leaf nodes and their parents: there are respectively one, two, three and four heavy edges (i.e. with $c(e) = 0.5$) from the internal nodes 1, 2, 3, and 4 to their children, i.e., the leaves in their subtree.

As shown in Fig. 2, our method adapts to the hierarchical structure with a change in the amount of compression from just one node to about three-fourths of the entire graph. We also show mean-

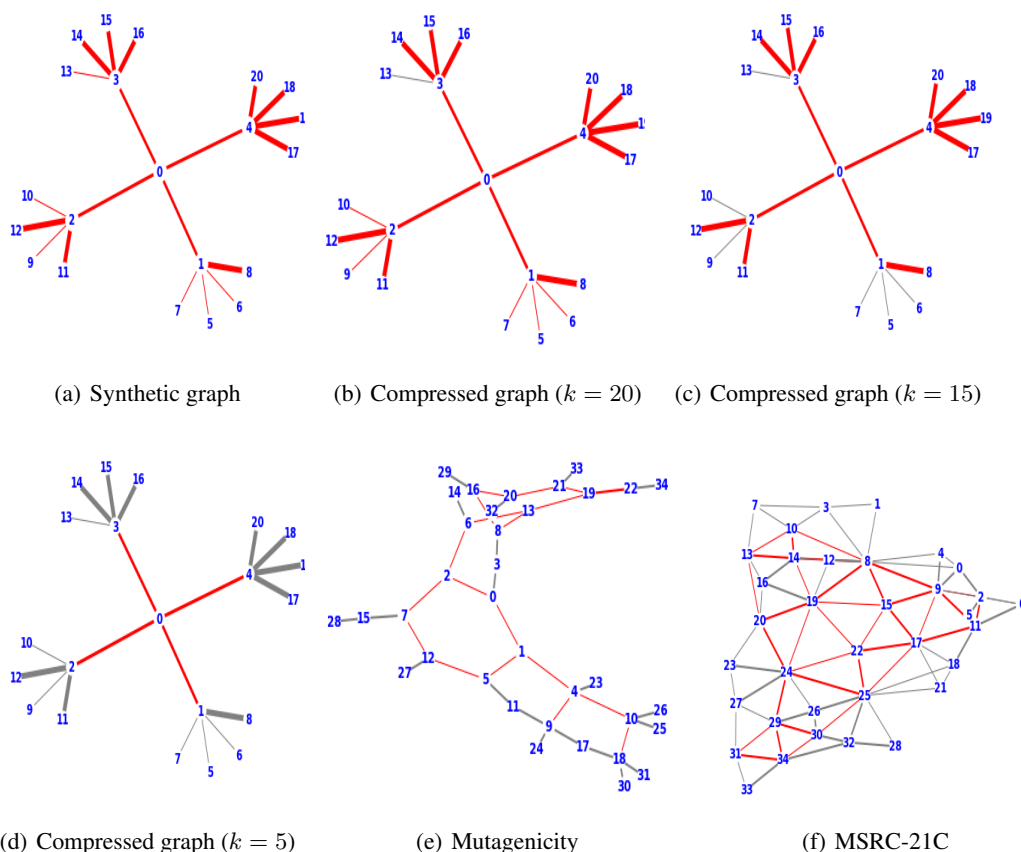

(a) Synthetic graph      (b) Compressed graph ($k = 20$)      (c) Compressed graph ($k = 15$)

(d) Compressed graph ($k = 5$)      (e) Mutagenicity      (f) MSRC-21C

Figure 2: **Visualizing compression on synthetic and real examples.** (a) A synthetic graph structured as a 4-ary tree of depth 2. The root 0 is connected to its neighbors by edges having $c(e) = 0.3$. All the other edges have either $c(e) = 0.5$ (thickest) or $c(e) = 0.1$ (lightest). The left out portions are shown in gray. (b) Leaf node 13, which is in the same subtree as three nodes with heavy edges 14-16, is the first to go. (c) Proceeding further, 9 and 10 are left out followed by the remaining nodes (i.e. 5, 6, 7) connected by light edges. (d) When the graph is compressed to 5 vertices, only the root and its neighbors remain despite bulkier subtrees, e.g. the one with node 4 and its neighbors, that are discarded. Thus, our method yields meaningful compression on this synthetic example. (e-f) Compressed structures pertaining to some sample graphs from real datasets that have some other interesting motifs. In each case, the compressed graph consists of red edges and the incident vertices, while the discarded parts are shown in gray. All the figures here are best viewed in color.

ingful compression on some examples from real datasets. The bottom row of Fig. 2 shows two such examples, one each from Mutagenicity and MSRC-21C. For these graphs, we used the same specification for $c$ as in section 4.1. The example from Mutagenicity contains patterns such as rings and backbone structures that are ubiquitous in molecules and proteins. Likewise, the other example is a good representative of the MSRC-21C dataset from computer vision.

Thus, our method encodes prior information, and provides fast and effective graph compression for downstream applications.

**Acknowledgments**

We thank the anonymous reviewers for their thoughtful questions that led to sections 3.3 and 3.4, and experiments on the Tox21 data. We are grateful to Andreas Loukas for the code of their algorithm [1]. VG and TJ were partially supported by a grant from the MIT-IBM collaboration.

## Footnotes

[1]https://tripod.nih.gov/tox21/challenge/data.jsp

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
