[Supplementary Material · GraphCompression_Final-13-17.pdf]

# A Supplementary Material

We provide here proofs of all the results from the main text.

**Proof of Theorem 1**

*Proof.* We will prove the result by contradiction.

Suppose there is some $\hat{e} \in E$ such that the optimal solution assigns $J^+(\hat{e}) > 0$ and $J^-(\hat{e}) > 0$. Then we note that

$$a \triangleq \min\{J^+(\hat{e}), J^-(\hat{e})\} > 0 \,.$$

Consider an alternate solution $(\tilde{J}^+, \tilde{J}^-)$ such that $\forall\, e \in E$,

$$\tilde{J}^+(e) = \begin{cases} J^+(e) - a & \text{if } e = \hat{e} \\ J^+(e) & \text{if } e \in E \setminus \{\hat{e}\} \end{cases},$$

$$\tilde{J}^-(e) = \begin{cases} J^-(e) - a & \text{if } e = \hat{e} \\ J^-(e) & \text{if } e \in E \setminus \{\hat{e}\} \end{cases}.$$

Clearly, $(\tilde{J}^+, \tilde{J}^-)$ is feasible for (1). Moreover, it achieves a lower value of the objective than the optimal solution $(J^+, J^-)$. Therefore, $(J^+, J^-)$ cannot be optimal. □

**Proof of Theorem 2**

*Proof.* We introduce non-negative Lagrangian vectors $\alpha$ and $\beta$, respectively, for the constraints $Ft \preceq c$ and $-c \preceq Ft$. We consider the terms in the objective that depend on $t$

$$g(t) = t^\top (\rho_1 - \rho_0) + \alpha^\top (Ft - c) - \beta^\top (Ft + c) \,.$$

The gradient $\nabla g(t)$ must vanish at optimality, so

$$\rho_1^* - \rho_0 + F^\top (\alpha - \beta) \;=\; \mathbf{0} \,.$$

The first part of the theorem follows immediately by defining $\eta = \beta - \alpha$. A closer look at (1) reveals that $\eta = J^- - J^+$ is, in fact, the net flow along the edges $e^-$ from (1).

Now, we prove the second part. By definition, in order for an edge $e$ to be active, at least one of $e^+$ and $e^-$ must be active, i.e., we must have $J^+(e) + J^-(e) > 0$. On the other hand, Theorem 1 implies that at least one of $J^+(e)$ and $J^-(e)$ is 0 for each $e \in E$ in the optimal solution. Combining these facts, we have that for any active edge $e$, exactly one of $e^+$ and $e^-$ is active, i.e., exactly one of the inequalities $J^+(e) > 0$ and $J^-(e) > 0$ must hold. This immediately implies, by complementary slackness, that exactly one of $\alpha(e)$ or $\beta(e)$ is 0. Thus, for any active edge $e$, either the lower bound or the upper bound on $Ft^*(e)$ in the constraints $-c(e) \le Ft^*(e) \le c(e)$ must become tight. Therefore, we must have $Ft^*(e) \in \{\pm c(e)\}$. □

**Proof of Theorem 3**

*Proof.* Note that since at least one coordinate of $\epsilon$ is strictly greater than 0, the feasible region is non-empty, and consequently, a unique projection exists. We introduce variables $\alpha \in \mathbb{R}$ and $\beta \in \mathbb{R}_+^d$, and form the Lagrangian

$$L(x, \alpha, \beta) \;=\; \frac{1}{2}\|x - y\|^2 - \alpha((x \odot \epsilon)^\top \mathbf{1} \,-\, 1) \,-\, \beta^\top (x \odot \epsilon) \,.$$

We now write the KKT conditions for the optimal solution $x$. For each $j \in [d]$, we must have

$$\begin{aligned} x_j - y_j - \alpha\epsilon_j - \beta_j\epsilon_j &= 0 \\ \epsilon_j x_j &\ge 0 \\ \beta_j &\ge 0 \\ \epsilon_j x_j \beta_j &= 0 \,. \end{aligned}$$

$$\text{Additionally,} \qquad \sum_{j=1}^d \epsilon_j x_j \;=\; 1 \,.$$

Clearly, for $j \in [d] \triangleq \{1, 2, \ldots, d\}, \epsilon_j = 0 \implies x_j = y_j$. Therefore, without loss of generality we assume in the rest of the proof that $\epsilon_j > 0$ for all $j$. Then we can immediately simplify the KKT conditions to

$$x_j - y_j - \alpha\epsilon_j - \beta_j\epsilon_j = 0 \tag{10}$$
$$x_j \geq 0 \tag{11}$$
$$\beta_j \geq 0 \ . \tag{12}$$
$$x_j\beta_j = 0 \tag{13}$$
$$\sum_{j=1}^{d} \epsilon_j x_j = 1 \tag{14}$$

We note that $x_j > 0 \overset{(13)}{\implies} \beta_j = 0 \overset{(10)}{\implies} y_j + \alpha\epsilon_j > 0 \overset{\epsilon_j > 0}{\implies} y_j/\epsilon_j > -\alpha$ , whereas

$$x_j = 0 \overset{(10)}{\implies} y_j + \alpha\epsilon_j = -\beta_j\epsilon_j \underset{\epsilon_j > 0}{\overset{(12)}{\implies}} y_j + \alpha\epsilon_j \leq 0 \overset{\epsilon_j > 0}{\implies} y_j/\epsilon_j \leq -\alpha \ .$$

This shows that the zero coordinates $x_j$ correspond to smaller values of $y_j/\epsilon_j$. Thus, we can sort the indices $j$ in non-increasing order based on the ratio $y_j/\epsilon_j$, reorder $x$ according to the sorted indices, and find an index $\ell \in [d]$ such that $x_j > 0$ for $j \in [\ell]$ and $0$ for $\ell < j \leq d$. Without loss of generality, we therefore assume that

$$x_1 \geq x_2 \ldots \geq x_\ell > 0 = x_{\ell+1} \ldots = x_d \ , \text{ and}$$
$$y_1/\epsilon_1 \geq y_2/\epsilon_2 \ldots \geq y_d/\epsilon_d \ . \tag{15}$$

We then have from (14) that

$$1 = \sum_{j=1}^{d} \epsilon_j x_j = \sum_{j=1}^{\ell} \epsilon_j x_j = \sum_{j=1}^{\ell} \epsilon_j (y_j + \alpha\epsilon_j)$$

$$\implies \alpha = \frac{1 - \sum_{j=1}^{\ell} \epsilon_j y_j}{\sum_{j=1}^{\ell} \epsilon_j^2} \ . \tag{16}$$

Thus, our task essentially boils down to finding the number of positive coordinates $\ell$. We now show that

$$\ell = \max \left\{ j \in [d] \ \middle| \ y_j + \epsilon_j \frac{(1 - \sum_{i=1}^{j} \epsilon_i y_i)}{\sum_{i=1}^{j} \epsilon_i^2} > 0 \right\} \ .$$

First consider $j < \ell$. Then $y_j/\epsilon_j > -\alpha$ for $j \in [\ell]$. Noting that $\epsilon_j > 0$ for all $j$ and using (16), we must have

$$y_j + \epsilon_j \frac{(1 - \sum_{i=1}^{j} \epsilon_i y_i)}{\sum_{i=1}^{j} \epsilon_i^2}$$

$$= \frac{\epsilon_j}{\sum_{i=1}^{j} \epsilon_i^2} \left( y_j \frac{\sum_{i=1}^{j} \epsilon_i^2}{\epsilon_j} + 1 - \sum_{i=1}^{j} \epsilon_i y_i \right)$$

which has the same sign as

$$y_j \frac{\sum_{i=1}^{j} \epsilon_i^2}{\epsilon_j} + 1 - \sum_{i=1}^{j} \epsilon_i y_i$$

$$= y_j \frac{\sum_{i=1}^{j} \epsilon_i^2}{\epsilon_j} + \sum_{i=j+1}^{\ell} \epsilon_i y_i + 1 - \sum_{i=1}^{\ell} \epsilon_i y_i$$

$$= y_j \frac{\sum_{i=1}^{j} \epsilon_i^2}{\epsilon_j} + \sum_{i=j+1}^{\ell} \epsilon_i y_i + \alpha \sum_{i=1}^{\ell} \epsilon_i^2$$

$$= \left( \frac{y_j}{\epsilon_j} + \alpha \right) \sum_{i=1}^{j} \epsilon_i^2 + \sum_{i=j+1}^{\ell} \epsilon_i^2 \left( \frac{y_i}{\epsilon_i} + \alpha \right)$$

$$> 0 \ .$$

Now consider $j = \ell$. Since $y_\ell/\epsilon_\ell > -\alpha$ and $\epsilon_\ell > 0$, we have $y_\ell + \alpha\epsilon_\ell > 0$. Thus

$$y_j + \epsilon_j \frac{(1 - \sum_{i=1}^{j} \epsilon_i y_i)}{\sum_{i=1}^{j} \epsilon_i^2} = y_\ell + \epsilon_\ell \frac{(1 - \sum_{i=1}^{\ell} \epsilon_i y_i)}{\sum_{i=1}^{\ell} \epsilon_i^2}$$

$$= y_\ell + \alpha\epsilon_\ell > 0 \,.$$

Finally, we consider $\ell < j \le d$. We note that

$$y_j + \epsilon_j \frac{(1 - \sum_{i=1}^{j} \epsilon_i y_i)}{\sum_{i=1}^{j} \epsilon_i^2}$$

$$= \frac{\epsilon_j}{\sum_{i=1}^{j} \epsilon_i^2} \left( y_j \frac{\sum_{i=1}^{j} \epsilon_i^2}{\epsilon_j} + 1 - \sum_{i=1}^{j} \epsilon_i y_i \right) \,,$$

which has the same sign as

$$y_j \frac{\sum_{i=1}^{j} \epsilon_i^2}{\epsilon_j} + 1 - \sum_{i=1}^{j} \epsilon_i y_i$$

$$= y_j \frac{\sum_{i=1}^{j} \epsilon_i^2}{\epsilon_j} + 1 - \sum_{i=1}^{\ell} \epsilon_i y_i - \sum_{i=\ell+1}^{j} \epsilon_i y_i$$

$$= y_j \frac{\sum_{i=1}^{j} \epsilon_i^2}{\epsilon_j} + \alpha \sum_{i=1}^{\ell} \epsilon_i^2 - \sum_{i=\ell+1}^{j} \epsilon_i y_i$$

$$= \left( \frac{y_j}{\epsilon_j} + \alpha \right) \sum_{i=1}^{\ell} \epsilon_i^2 + \sum_{i=\ell+1}^{j} \epsilon_i^2 \left( \frac{y_j}{\epsilon_j} - \frac{y_i}{\epsilon_i} \right) \,,$$

$$\le 0 \,,$$

by leveraging the sorted property in (15) and the fact that $y_j/\epsilon_j \le -\alpha$ for $j \in [\ell]$.

Therefore, we have shown that $y_j + \epsilon_j \frac{(1 - \sum_{i=1}^{j} \epsilon_i y_i)}{\sum_{i=1}^{j} \epsilon_i^2} > 0$ for all $j \in [\ell]$, and at most 0 for $\ell < j \le d$. Algorithm 1 implements this procedure, and that proves its correctness. The $O(d \log d)$ time complexity is due to the cost of sorting the indices $j \in [d]$ based on $y_j/\epsilon_j$. $\qquad\square$

**Proof of Theorem 4**

*Proof.* Recall the formulation (6):

$$\min_{\substack{\epsilon \in \mathcal{E}_k \\ \tilde{\rho}_1 \odot \epsilon \in \Delta(V)}} \max_{t \in \mathcal{T}_{F,c}} \underbrace{t^\top (\tilde{\rho}_1 \odot \epsilon - \rho_0) + \frac{\lambda}{2} ||\tilde{\rho}_1||^2}_{\phi(\epsilon, t, \tilde{\rho}_1)} \,.$$

Making the constraints $\mathcal{E}_k$ explicit, we get

$$\min_{\substack{\epsilon \in \{0,1\}^{|V|} \\ \epsilon^\top \mathbf{1} \le k}} \left( \min_{\substack{\tilde{\rho}_1 \in \mathbb{R}^{|V|} \\ \tilde{\rho}_1 \odot \epsilon \in \Delta(V)}} \max_{t \in \mathcal{T}_{F,c}} \phi(\epsilon, t, \tilde{\rho}_1) \right) \,. \tag{17}$$

Note that for any fixed $\epsilon$ (a) $\{\tilde{\rho}_1 \in \mathbb{R}^{|V|} \mid (\tilde{\rho}_1 \odot \epsilon) \in \Delta(V)\}$ is convex, and $\mathcal{T}_{F,c}$ is convex and compact, (b) $\phi(\epsilon, t, \tilde{\rho}_1)$ is continuous, and (c) for every fixed $t$, $\phi(\epsilon, t, \cdot)$ is convex in $\tilde{\rho}_1$; while for every fixed $\tilde{\rho}_1$, $\phi(\epsilon, \cdot, \tilde{\rho}_1)$ is linear (thus concave) in $t$. Therefore, invoking the Sion's minimax theorem [60], we can swap the order of $\min$ and $\max$ within the parentheses in (17), and obtain

$$\min_{\substack{\epsilon \in \{0,1\}^{|V|} \\ \epsilon^\top \mathbf{1} \le k}} \left( \max_{t \in \mathcal{T}_{F,c}} \min_{\substack{\tilde{\rho}_1 \in \mathbb{R}^{|V|} \\ \tilde{\rho}_1 \odot \epsilon \in \Delta(V)}} \phi(\epsilon, t, \tilde{\rho}_1) \right) \,. \tag{18}$$

We introduce Lagrangian variables $\nu \in \mathbb{R}_+^{|V|}$ and $\zeta \in \mathbb{R}$, respectively, for the simplex constraints (a) $\mathbf{0} \preceq \tilde{\rho}_1 \odot \epsilon$ and (b) $(\tilde{\rho}_1 \odot \epsilon)^\top \mathbf{1} = 1$ to get

$$\Phi(\epsilon, t, \tilde{\rho}_1, \nu, \zeta) \triangleq \phi(\epsilon, t, \tilde{\rho}_1) + \zeta((\tilde{\rho}_1 \odot \epsilon)^\top \mathbf{1} - 1) - \nu^\top (\tilde{\rho}_1 \odot \epsilon) . \qquad (19)$$

Applying the optimality conditions, we note for any fixed pair $(\epsilon, t)$, the corresponding optimal $\tilde{\rho}_1$ must satisfy

$$\partial_{\tilde{\rho}_1} \Phi(\epsilon, t, \tilde{\rho}_1, \nu, \zeta) = 0 ,$$

whereby

$$\tilde{\rho}_1 = -\left( \frac{(t - \nu) \odot \epsilon + \zeta\epsilon}{\lambda} \right) . \qquad (20)$$

Plugging in $\tilde{\rho}_1$ from (20) into (19), we thus have the following equivalent dual formulation for (18)

$$\min_{\substack{\epsilon \in \{0,1\}^{|V|} \\ \epsilon^\top \mathbf{1} \leq k}} \max_{t \in \mathcal{T}_{F,c}} \max_{\nu \in \mathbb{R}_+^{|V|}, \zeta \in \mathbb{R}} \mathcal{M}(\epsilon, t, \nu, \zeta) ,$$

where

$$\mathcal{M}(\epsilon, t, \nu, \zeta) = \frac{-1}{2\lambda} ||(t - \nu + \zeta\mathbf{1}) \odot \epsilon||^2 - t^\top \rho_0 - \zeta . \qquad (21)$$

Invoking the first order convex optimality condition for constrained optimization, the $\hat{\epsilon}$ obtained from relaxation of (6) is optimal if and only if

$$\mathbf{0} \in \left\{ \partial_\epsilon \underbrace{\max_{t \in \mathcal{T}_{F,c}} \max_{\nu \in \mathbb{R}_+^{|V|}, \zeta \in \mathbb{R}} \mathcal{M}(\epsilon, t, \nu, \zeta)}_{(A)} + \mathbb{N} \right\}, \qquad (22)$$

where $\mathbb{N}$ is the normal cone of the relaxed constraints

$$\tilde{\mathcal{E}}_k = \left\{ \epsilon \in [0,1]^{|V|} \,\middle|\, \epsilon^\top \mathbf{1} \leq k \right\} .$$

Now we note that for any vector $x$, we can write $x \odot \epsilon = D(\epsilon)x$, where $D(\epsilon)$ is the diagonal matrix corresponding to $\epsilon$. Also, since $\epsilon \in \{0,1\}^{|V|}$, we get $D(\epsilon)^\top D(\epsilon) = D(\epsilon)$. Thus, for any $x$, we have

$$||x \odot \epsilon||^2 = ||D(\epsilon)x||^2 = (D(\epsilon)x)^\top D(\epsilon)x = x^\top D(\epsilon)^\top D(\epsilon)x = x^\top D(\epsilon)x .$$

In particular, we can simplify $||(t - \nu + \zeta\mathbf{1}) \odot \epsilon||^2$ in (21) when we set $x$ to $t - \nu + \zeta\mathbf{1}$. The theorem statement then follows immediately from (22) by representing $\mathbb{N}$ at the integral point $\epsilon^*$ and leveraging the non-negative dual parameter associated with the constraint $\epsilon^\top \mathbf{1} \leq k$. $\qquad \square$

**Proof of Theorem 5**

*Proof.* We will use the shorthand $\tilde{\rho}_{1v}$ for $\tilde{\rho}_1(v)$, and likewise for indexing $t$, $\nu$, and $\epsilon$. Using (20),

$$\tilde{\rho}_1 = -\left( \frac{(t - \nu) \odot \epsilon + \zeta\epsilon}{\lambda} \right) = -\left( \frac{t - \nu + \zeta\mathbf{1}}{\lambda} \right) \odot \epsilon .$$

Therefore,

$$\tilde{\rho}_1 \odot \epsilon = -\left( \frac{t - \nu + \zeta\mathbf{1}}{\lambda} \right) \odot \epsilon \odot \epsilon = `-\left( \frac{t - \nu + \zeta\mathbf{1}}{\lambda} \right) \odot \epsilon = \tilde{\rho}_1 , \qquad (23)$$

since $\epsilon \in \{0,1\}^{|V|}$. We write one of the KKT conditions for optimality

$$\tilde{\rho}_1 \odot \epsilon \odot \nu = \tilde{\rho}_1 \odot \nu = \mathbf{0} .$$

We consider the different cases. Note that for $v \in V$, using (23), we have

$$\tilde{\rho}_{1v} > 0 \implies \nu_v = 0 \implies \tilde{\rho}_{1v} = -\frac{(t_v + \zeta)\epsilon_v}{\lambda} , \qquad (24)$$

and
$$\tilde{\rho}_{1v} = 0 \implies (t_v - \nu_v + \zeta)\epsilon_v = 0 \implies \nu_v\epsilon_v = (t_v + \zeta)\epsilon_v . \tag{25}$$

But since $\nu_v \geq 0$, and $\epsilon_v \in \{0, 1\}$, we note that $\nu_v\epsilon_v \geq 0$. Then, by (25), we have

$$\nu_v\epsilon_v \geq 0 \implies -\frac{(t_v + \zeta)\epsilon_v}{\lambda} \leq 0 = \tilde{\rho}_{1v} . \tag{26}$$

Combining (24) and (26), we can write

$$\tilde{\rho}_{1v} = \max\left\{-\frac{(t_v + \zeta)\epsilon_v}{\lambda}, \ 0\right\} = \frac{\epsilon_v}{\lambda}\max\left\{-(t_v + \zeta), \ 0\right\} , \tag{27}$$

since $\epsilon_v \geq 0$ for all $v \in V$ and $\lambda > 0$. Therefore, we get $\tilde{\rho}_1 = \dfrac{\epsilon}{\lambda} \odot r_+$, where $r = -(t + \zeta\mathbf{1})$, and $r_+$ is computed by setting the negative coordinates of $r$ to 0.

Moreover, since $\tilde{\rho}_1 \odot \nu = \mathbf{0}$, we can eliminate $\nu$ from (19) and write (17) as

$$\min_{\epsilon\in\mathcal{E}_k} \ \max_{t\in\mathcal{T}_{F,c}} \max_{\zeta\in\mathbb{R}} \quad t^\top(\tilde{\rho}_1 - \rho_0) + \frac{\lambda}{2}||\tilde{\rho}_1||^2 + \zeta(\tilde{\rho}_1^\top\mathbf{1} - 1)$$

Substituting for $\tilde{\rho}_1$ from (27), we obtain the following equivalent problem

$$\min_{\epsilon\in\mathcal{E}_k} \ \max_{t\in\mathcal{T}_{F,c}} \max_{\zeta\in\mathbb{R}} \quad -\frac{r^\top}{\lambda}(\epsilon \odot r_+) + \frac{1}{2\lambda}r_+^\top(\epsilon \odot r_+) - t^\top\rho_0 - \zeta ,$$

$$= \min_{\epsilon\in\mathcal{E}_k} \ \max_{t\in\mathcal{T}_{F,c}} \max_{\zeta\in\mathbb{R}} \quad \frac{-1}{2\lambda}\epsilon^\top(r_+ \odot (2r - r_+)) - t^\top\rho_0 - \zeta ,$$

which can be written as

$$\min_{\epsilon\in\mathcal{E}_k} \ \max_{t\in\mathcal{T}_{F,c}} \max_{\zeta\in\mathbb{R}} \quad -\frac{1}{2\lambda}\sum_{v:r_v\geq 0}\epsilon_v r_v^2 - t^\top\rho_0 - \zeta$$

$$= \min_{\epsilon\in\mathcal{E}_k} \ \max_{t\in\mathcal{T}_{F,c}} \max_{\zeta\in\mathbb{R}} \quad -\frac{1}{2\lambda}\sum_{v:t_v\leq -\zeta}\epsilon_v r_v^2 - t^\top\rho_0 - \zeta$$

$$= \min_{\epsilon\in\mathcal{E}_k} \ \max_{t\in\mathcal{T}_{F,c}} \max_{\zeta\in\mathbb{R}} \quad -\frac{1}{2\lambda}\sum_{v:t_v\leq -\zeta}\epsilon_v(t_v + \zeta)^2 - t^\top\rho_0 - \zeta$$

$$= \min_{\epsilon\in\mathcal{E}_k} \ \max_{t\in\mathcal{T}_{F,c}} \max_{\zeta\in\mathbb{R}} \quad -\frac{1}{2\lambda}\sum_{v:t_v\leq -\zeta}\left(\epsilon_v(t_v + \zeta)^2 + 2\lambda t_v\rho_{0,v}\right) - \sum_{v:t_v> -\zeta}t_v\rho_{0,v} - \zeta .$$

$\square$