[Reviews · NeurIPS 2019]

Reviewer 1



Update after author response The authors have addressed my concerns regarding learning the costs, and unsupervised costs associated with degrees. I'd like to see these formulations in the final version. The paper's contribution includes both the usage of OT for graph compression and the specific optimization based on Boolean relaxation. For the latter, it may be beneficial to compare with a greedy Frank-Wolfe approach to handle both the support cardinality constraint and the simplex constraint. The revision could comment on the feasibility of this including a full line search and the fully-corrective variant of Frank-Wolfe. _______________________ Overall this is a good paper which solves an interesting problem in a novel but principled way, and along the way solves a general subproblem. It seems to be an original work, with clear significance to certain forms of network analysis. With regard to downstream tasks, there seems to be a heuristic step to set the costs appropriately to preserve the information. Can a more explicit linkage to the costs be made if the training labels are available? Or is this simply not feasible. In some respects, the experiments are a bit of a straw man since the cost function is defined in supervised terms of node labels, which I assume are related to the graph label. In practice the user is left with this choice. Although learning the cost is mentioned (Lines 202–207), how would this be done in practice (as an outer loop), would it scale, would it overfit? Is there a way to relate the optimal transport to other compression techniques if the costs are defined in a manner that does not consider extrinsic information, say using only the node degree, path length, etc. Minor points for clarity: Clarify what is meant by features of a graph at line 15. Line 45 "0" -> "zero" Line 48, the assumptions are not clear in context. Line 62, Clarify that mixed graphs contain both directed and undirected edges. It would be useful to have the baseline performance measure of the uncompressed graph in Table 1. Line 181, Clearly explain the relationship between node labels and graph labels in the experimental data section. Explain a bit more in text the Laplacian quadratic reference at line 22. Remove parentheses around bracketed references, especially in introduction.

Reviewer 2



This paper proposes a new approach for graph compression by optimal transport (OT). Specifically, it adds a l_0 norm constraint to the LP programming induced by OT and relaxes it to the projection into a simplex. The final problem can be solved by a customized mirror prox method. Empirical studies on are also shown. Following are my concerns: 1. Theorem 4 presents a condition under which the relaxation can recover the optimal solution of the original problem. It seems better to give some more discussions, .e.g., the mildness of the requirements. 2. Figure 1 claims the high efficiency of the proposed method, but the experiment settings and the implementation details are missing, so whether the comparisons are fair is not clear.

Reviewer 3



I regard the contribution as novel and exciting. First of all, from the methodological perspective. I appreciate the clarity and high technical quality of the submission. Nevertheless, I have a few technical questions/concerns: 1. Experimental setup, and Table 1. All experiments were provided for sufficiently small and very sparse graphs. For better understanding the value and the limits of the approach, it would be great to have more experiments over larger datasets. If larger datasets are available, do you have experiments with a higher compression ratio? What are the results? 2. For DHFR dataset, the results of different methods are almost the same. Do you have any insight regarding that? Is there anything unique in this dataset? Update after the authors' response: Many thanks to the authors for addressing my concerns. I have no more concerns regarding the paper.

[Author Response · NeurIPS 2019]

We are grateful to the reviewers for their feedback. We address their concerns here.

**Reviewer #1:** Thank you very much for your thoughtful and detailed review. We will include all your suggestions.

(1) **Learning costs from label information**. We discuss a principled way while ensuring the resulting formulation is
efficiently solvable. Define $\ell(i,j) = 1$ for edge $(i,j)$ if vertices $i$ and $j$ have the same label, and $-1$ otherwise. Let the
cost function be parameterized by $\theta = (\theta_s, \theta_d), \theta_s > 0, \theta_d > 0$. Define $c_\theta(i,j) = 0.5(\theta_d(1 - \ell(i,j)) + \theta_s(1 + \ell(i,j)))$.
Thus, $c_\theta(i,j) \in \{\theta_s, \theta_d\}$ depending on whether $i$ and $j$ have the same label. We now extend eq. 3 in paper to have

$$\min_{\mathbf{0} \prec \theta} \min_{\rho_1 \in \Delta(V), ||\rho_1||_0 \leq k} \max_{t \in \mathbb{R}^{|V|}, -c_\theta \preceq Ft \preceq c_\theta} t^\top(\rho_1 - \rho_0) + 0.5\lambda||\rho_1||^2 + 0.5||\theta||_2^2 .$$

Proceeding as in the paper, we obtain the following equivalent formulation (using $\psi$ from eq. 9 in Theorem 5):

$$\min_{\mathbf{0} \prec \theta} \min_{\epsilon \in \mathcal{E}_k} \max_{\zeta \in \mathbb{R}} \max_{t \in \mathbb{R}^{|V|}, -c_\theta \preceq Ft \preceq c_\theta} \psi_{\rho_0}(\epsilon, t, \zeta) + 0.5||\theta||_2^2 .$$

We can thus efficiently solve the convex-concave problem obtained by relaxing each coordinate of $\epsilon$ to $[0,1]$ in

$$\min_{\epsilon \in \mathcal{E}_k} \min_{\mathbf{0} \prec \theta, \mathbf{0} \preceq \alpha, \ \mathbf{0} \preceq \beta} \max_{\zeta \in \mathbb{R}, t \in \mathbb{R}^{|V|}} \psi_{\rho_0}(\epsilon, t, \zeta) + 0.5||\theta||_2^2 + \alpha^\top(Ft - c_\theta) - \beta^\top(Ft + c_\theta) .$$

(2) **Relation to other compression techniques**. Most successful algorithms try to preserve the graph spectrum via a
multi-level coarsening procedure: at each level they compute a matching of vertices and merge the matched vertices, e.g.,
Heavy Edge contracts those edges $(i,j)$ that are incident on low degree vertices. Likewise REC follows a randomized
greedy procedure for generating maximal matching incrementally. If we set the cost $c(i,j) = \max(d_i, d_j)$ in our OTC
framework, we will incentivize flow on edges with low degree vertices, and in turn, compression of one of their end
points. The vertices not in support of target distribution $\rho_1$ may then be viewed as being matched to (a subset of)
adjacent vertices that they transfer flow to. Unlike other methods, our approach is (a) flexible in terms of defining
$c(i,j)$, and (b) not greedy, therefore, less susceptible to errors inherent in iterative greedy matching procedures.

**Reviewer #2:** Thank you very much for your constructive feedback. We address both your concerns here.

(1) **Discussion of assumption in Theorem 4**. The quantity $|\hat{t}(v) - \hat{\nu}(v) + \hat{\zeta}|$ in eq. 7 may be viewed as the strength
of a signal. Then, we require the vertices in support of optimal $\rho_1$ to have a strictly higher signal that the vertices
not in the support. Such signal detection conditions appear in various contexts and often have information theoretic
implications, e.g., Ising models (Santhanam and Wainwright, *IEEE Transactions on Information Theory*, 2012). Eq. 7 is
also reminiscent of the $\beta$-min condition on regression coefficients for variable selection with Lasso in high-dimensional
linear models (Bühlmann, *Bernoulli*, 2012), however, we do not require analogs for stringent assumptions required
by Lasso such as restricted eigenvalue, and thus our Boolean relaxations are preferable to solving an $\ell_1$-regularized
problem. We will include this discussion based on your feedback.
(2) **Experiment settings for Fig. 1**. We apologize for the confusion. The setup for Fig. 1 is identical to that for Table 1.
Fig. 1 provides visualization of compression times in addition to accuracy results from Table 1. To ensure the fairness
of our experiments, for each dataset, we first executed the randomized algorithm (REC) over each graph, and then set
the target number of nodes for other methods to the number of vertices in corresponding compressed graphs from REC.
We performed 5 such independent executions of REC to mitigate the effect of randomness. Likewise, the compressed
graphs were partitioned into multiple train-test sets for each of the different fractions, and average accuracy and standard
deviations along with compression times were plotted (please see section 4.1 for the details of our implementation).

**Reviewer #3:** Many thanks for a comprehensive review. We are glad that you find our work exciting.
(1) **Scalability of our approach**. Please note that almost all state-of-the-art compression methods are compute-intensive
since they need to perform matching as a sub-step at multiple levels. In contrast, there are no major computational
bottlenecks in our approach. In particular, please note that both the projection steps in Algorithm 2 can be solved
efficiently by existing algorithms. In Algorithm 1, we perform sorting that requires $O(|V| \log |V|)$ time. We believe
that even this log factor may be removed from our computation by a more sophisticated algorithm, much like (Condat
[43]) managed to improve on the algorithm for Euclidean projection on the simplex by (Duchi et al. [41]).
We also experimented with the Tox21_ARLBD data (https://tripod.nih.gov/tox21/challenge/data.jsp), which consists
of 8589 graphs, based on your suggestion. Both our method and Algebraic distance performed very well in terms of
classification accuracy ($\sim 97\%$) on this data. Our approach took in total about 39 seconds to compress graphs in this
data to 90% (low compression), and about 41 seconds in total to compress to 10% (high compression). In contrast, the
Algebraic distance method took about 48 seconds to compress to 90% and a significantly longer time, i.e., 3.5 minutes
to compress to 10%. The other methods failed to compress this data. Please note that Algebraic distance is amongst the
fastest state-of-the-art compression methods (Chen and Safro [13]).
(2) **Regarding DHFR dataset**. Indeed, though OTC performs the best on DHFR, the performance of most methods is
similar (except REC, which lags behind). In contrast, REC performs better than all methods except OTC on MSRC-9.
This seems to suggest that REC performs well on graphs with strong connectivity, while others might be better on data
with a long backbone besides these ring structures. We believe robust performance of OTC across datasets comprising
graphs with vastly different topologies underscores the promise of our approach.

[Meta-Review · NeurIPS 2019]

This paper proposes a new approach for graph compression by optimal transport. It adds a sparsity constraint to the resulting Linear Program and relaxes it to the projection into a simplex. The final problem can be solved by a customized mirror prox method. Empirical studies on are also shown. All reviewers agreed that this is a nicely written paper that studies an original problem with a principled approach. This problem is likely to have applications to the broad ML community and therefore the AC recommends acceptance.